# Examination of Respiratory Disturbance Index Before and After Cheiloplasty and Palatoplasty

**DOI:** 10.3390/diseases13030064

**Published:** 2025-02-21

**Authors:** Ryo Murasugi, Hitoshi Kawanabe, Ayano Murakami, Yasuhiko Fukuya, Hideto Imura, Nagato Natsume, Ken Sato, Seiko Mitachi, Kazunori Fukui

**Affiliations:** 1Division of Orthodontics and Dentofacial Orthopedics, Department of Oral Growth and Development, Ohu University School of Dentistry, Fukushima 963-8041, Japan; muraryo320@gmail.com; 2Department of Dentofacial Orthopedics, Graduate School of Dentistry, Ohu University School of Dentistry, Fukushima 963-8041, Japan; 3Department of Plastic and Reconstructive Surgery, Ohtanishinouchi Hospital, Ohta General Hospital Foundation, Fukushima 963-8558, Japan; 4Cleft Lip and Palate Center, Aichi Gakuin University Dental Hospital, Nagoya 464-8651, Japan; 5School of Dentistry, Ohu University, Fukushima 963-8041, Japan; 6School of Computer Science, Tokyo University of Technology, Tokyo 192-0982, Japan

**Keywords:** pediatric sleep-disordered breathing, respiratory disturbance index, cleft lip and palate, infants, out-of-center sleep testing, cheiloplasty, palatoplasty, sleep apnea, Fiber-Based Sleep Apnea Syndrome Sensor^®^

## Abstract

**Background/Objectives:** Pediatric sleep-disordered breathing (SDB) can cause behavioral and cognitive problems and even physical growth impairment, but it is often under-recognized. Cleft lip and/or palate (CLP) is a common birth defect and known risk factor for SDB. In this study, we examined the sleep breathing status in infants with unilateral CLP (UCLP) before and after cheiloplasty and palatoplasty. **Methods:** This prospective before–after study included infants with UCLP who could undergo the sleep breathing test and sleep for >1 h. Their sleep breathing status was assessed using a fiber-based sleep apnea sensor (Fiber-Based Sleep Apnea Syndrome Sensor^®^) on the day before surgery and 1–3 d after surgery. We calculated and compared the pre- and postoperative respiratory disturbance index (RDI) following the criteria proposed by the American Academy of Sleep Medicine. **Results:** The mean RDI significantly improved both after cheiloplasty (from 7.5 ± 4.6 to 2.7 ± 1.4 events/h, *p* = 0.007) and after palatoplasty (from 4.4 ± 2.3 to 1.7 ± 0.4 events/h, *p* = 0.010). **Conclusions:** Cheiloplasty and palatoplasty could improve SDB and reduce its adverse effects on the physical growth and development of infants with UCLP.

## 1. Introduction

Pediatric sleep-disordered breathing (SDB), a term proposed by Guilleminault [1] in 1976, was first defined as an independent disease in the second edition of the International Classification of Sleep Disorders (ICSD-2) in 2005 [2]. According to the 2014 ICSD-3 criteria [3], pediatric SDB is characterized by daytime sleepiness, attention deficit, hyperactivity, moodiness, irritability, and learning and other cognitive problems, such as poor academic performance [4,5]. Furthermore, the sleep impairment caused by SDB results in the inhibition of growth hormone secretion, which may cause physical growth impairment in children [6]. Moreover, given the respiratory distress during sleep, SDB can induce sudden infant death syndrome [7]. Therefore, the early detection of pediatric SDB is crucial to prevent these adverse effects.

The American Academy of Pediatrics recommends the use of polysomnography (PSG) for the diagnosis of SDB in children [8] and adults [2]. In addition to the international diagnostic standard, the apnea–hypopnea index (AHI), criteria for the diagnosis of pediatric SDB in the ICSD-3 [3] include the presence of snoring, gasping breathing, respiratory arrest, abnormal posture, excessive body movement, excessive perspiration, and nocturia. In 2019, Ameet et al. described PSG findings in infants [9]. However, PSG is a multichannel procedure for monitoring multiple body functions and should be performed in specialized facilities. Owing to resource constraints, PSG often cannot be performed, and only a few facilities in Japan perform it in infants [10]. Recently, out-of-center sleep testing (OCST), which allows testing at home using noninvasive and nonrestrictive devices [11], has increasingly been used and is considered very useful. OCST allows the identification of infants with potential SDB who can subsequently undergo PSG at specialized facilities for definitive diagnosis.

Cleft lip and/or palate (CLP) is the most common birth defect in Japan, with unilateral defects being 5.4 times more prevalent than bilateral ones [12]. CLP is a known risk factor for SDB. Preschool children with CLP have a five times greater risk of obstructive sleep apnea compared with children without CLP. Nonetheless, obstructive sleep apnea appears to be under-recognized in this population [13,14].

In this study, we aimed to compare the sleep breathing status, as evaluated by the respiratory disturbance index (RDI), before and after cheiloplasty and/or palatoplasty in infants with unilateral CLP (UCLP) and investigate its effect on sleep.

## 2. Materials and Methods

### 2.1. Participants and Procedure

This prospective before–after study included infants who underwent cheiloplasty and/or palatoplasty for UCLP between 2017 and 2020 at the Department of Orthodontics of Ohu University Dental Hospital; Department of Plastic Surgery of Ohta Nishinouchi Hospital, Ohta Foundation; and Cleft Lip and Palate Center of Aichi Gakuin University Dental Hospital. Infants who could not undergo sleep breathing tests due to body movement or other reasons, those with other craniofacial syndromes in addition to UCLP, and those with a sleeping time < 1 h were excluded.

During the study period, cheiloplasty was performed at Ohta Foundation Ohta Nishinouchi Hospital and Aichi Gakuin University Dental Hospital, while palatoplasty was performed at the Aichi Gakuin University Dental Hospital. Cheiloplasty and palatoplasty were limited to the Millard method and pushback method, respectively. All surgeons had similar experience levels.

This study was approved by the Ohu University Ethics Review Board (No. 198), Ohta Foundation Ohta Nishinouchi Hospital Ethics Review Board (No. 45), and Aichi Gakuin University Dental Hospital Ethics Review Board (No. 611). The parents of eligible infants provided written informed consent after the OCST procedure and the study’s purpose and methods were explained to them.

### 2.2. OCST

#### 2.2.1. Testing Method

The participants’ sleep breathing status was evaluated overnight at the hospital on the day before surgery and 1–3 d after surgery using an optical fiber-based sleep apnea sensor (Fiber-Based Sleep Apnea Syndrome Sensor^®^; hereinafter F-SAS Sensor^®^; AI Coabi (Academy & Industry Cohabitation), Mito-city, Japan).

The F-SAS Sensor^®^ is used for noncontact OCST and comprises a sleep apnea sensor, which is a quartz optical fiber sheet that responds to the amplitude pressure of the chest during breathing, connected to a control unit with an SC connector. The control unit comprises a power supply (DC12V), light source (red light emitting diode = 650 nm), light receiver (optical power meter), signal control/recording part, display, and SD memory card (Figure 1). The main power supply uses the household power supply of AC100V.

The quartz optical fiber sheet was laid on the infant’s bed in the hospital room. The infant was put to sleep as usual. The measurement start/end button on the control unit was pressed to measure the infant’s breathing status during sleep and again upon waking up to end the measurement. The overnight measurement data were stored on the SD card in the control unit. The frequency of the measured data was converted to the time axis through Fourier transformation (Figure 2). The results were presented as the RDI, which was measured by the F-SAS Sensor^®^ on an hourly basis [15].

#### 2.2.2. Evaluation Method

A typical waveform during stable breathing after falling asleep was extracted and used to determine the normal breathing pattern. Specifically, the duration of one breath was used as a reference value, and the amplitude pressure of the chest for two or more breaths was used as the baseline. Apnea was defined as sensor signal flattening for two or more breaths and hypopnea as a signal decrease of ≥50% (Figure 3).

The duration of one breath (s) was calculated from the stable breathing waveform and multiplied by two to determine the threshold for diagnosing apnea. As shown in Figure 3, the duration of one breath was determined as 2.3 s, and infants with a flat baseline for at least 4.6 s (two breaths) were diagnosed with apnea.

The RDI was calculated by dividing the total number of apnea and hypopnea events during the test by the total test time. SDB severity was classified using the RDI as follows: RDI < 5, normal breathing; RDI 5–14, mild; RDI 15–29, moderate; and RDI ≥ 30, severe SDB (Table 1). The sleep evaluation was performed using the criteria of the American Academy of Sleep Medicine [16].

### 2.3. Statistical Analysis

Statistical analyses were performed using IBM SPSS Statistics 24.0 (IBM Corp., Armonk, NY, USA). The paired *t*-test was used for comparisons of the RDI before and after surgery. Statistical significance was set at *p* > 0.05.

## 3. Results

### 3.1. Participants’ Characteristics

Among 22 infants with UCLP, 12 (9 boys and 3 girls) were included in the study. Among them, six infants underwent cheiloplasty (5 boys and 1 girl; mean age: 6.0 ± 1.9 months) and six underwent palatoplasty (4 boys and 2 girls; average age: 20.0 ± 4.2 months) (Table 2).

### 3.2. Sleep Breathing Status Before and After Surgery

Compared with preoperative values, the mean RDI showed a significant reduction postoperatively both for cheiloplasty (7.5 vs. 2.7 events/h, *p* = 0.007; Table 3 and Figure 4) and palatoplasty (4.4 vs. 1.7 events/h, *p* = 0.010; Table 3 and Figure 5). This indicated postoperative improvement in the sleep breathing status.

## 4. Discussion

In this before–after study, using OCST, we determined that infants with UCLP demonstrated mild SDB and that corrective surgery (cheiloplasty and palatoplasty) significantly improved their sleep breathing status, as measured by the RDI.

In 2019, Matlen et al. used PSG to examine infants with pediatric SDB, including infants with cleft palate [17]. Although the severity classification of pediatric SDB is based on the ICSD-3 [3], the diagnostic criteria for infants remain unclear. In the ICSD-3, pediatric SDB is defined based on the AHI calculated through PSG. However, applying the standard for abnormal sleep breathing status (AHI > 1 events/h) could result in many infants being erroneously diagnosed with SDB [9]. An erroneous diagnosis may lead to the initiation of drug control, including the administration of caffeine or theophylline, and treatment using continuous positive airway pressure. This causes body stress on the infant and may induce adverse effects, including drug reactions [18]. Therefore, establishing diagnostic criteria specific to infants and children is imperative.

PSG is recommended for the diagnosis of pediatric and adult SDB. This method provides a detailed examination using numerous channels, including an electroencephalogram, eye movement, an electrocardiogram, a pulse oximeter, a chin electromyogram, a lower limb electromyogram, a thoracoabdominal belt, and airflow, breathing, position, and snoring sensors. These allow the determination of the sleep onset/wake time and sleep states, including sleep depth and sleep cycle. However, PSG has numerous drawbacks for infants, including skin problems, pain, and stress due to electrode attachment/detachment [10].

In the current study, we applied OCST using the F-SAS Sensor^®^, a simple sleep test device that is used by laying a sheet on the bed without attaching anything to the infant’s body, allowing the infant freedom to move or turn over [15]. Therefore, unlike PSG, the F-SAS Sensor^®^ is a noninvasive and nonrestrictive screening device that allows simple tests to be conducted without stressing infants. In addition, Mitachi et al. demonstrated that PLSX data and PSG (Alice5) data are well correlated with F-SAS Sensor^®^ data [11].

### 4.1. Relationship of CLP with Sleep Apnea

In infants with UCLP, the muscles around the lips (e.g., orbicularis oris, nasalis muscle, and levator labii) divided by the cleft cause persistent unbalanced tension to the nasal cartilage, septum, and upper lip. This results in morphological soft tissue and bone/cartilage abnormalities [19,20], including the deviation of the nasal septum and compensatory hypertrophy of the inferior nasal concha [21]. In the cleft part of the palate, a part of the levator veli palatini and the longitudinal section of the palatopharyngeus muscle adhere to the posterior margin of the palatine bone. This results in the aplasia and hypoplasia of the palatal muscles and levator veli palatini, respectively, and the continuation of the nasal and oral cavities. In normal nasal breathing, during inspiration, a negative pressure is created in the thoracic and oral cavities, which becomes positive during expiration [22]. Furthermore, air taken in through nasal breathing is heated and humidified in the nasal cavity before passing through the respiratory tract. In infants with CLP, owing to the continuous nasal and oral cavity, oral rather than nasal breathing is the main activity. Moreover, dry cold air passes through the airways since the nasal cavity cannot heat or humidify it. Additionally, infants with UCLP have a unilateral upper-lip cleft. Inside the cleft, the apex of the cupid arch on the cleft side is displaced upward. In front of the cleft, the ridge of the mucous membrane of the lip mucocutaneous junction disappears. At the external nose, the nose tip and bridge are deviated to the non-cleft side, the nasal ala and external nostril are flattened, and the base of the nasal ala is outwardly and downwardly deviated on the cleft side. In the cleft palate, the alveolar stump inside the cleft has a forward protrusion, which shows a difference from the alveolar stump outside the cleft. Hence, in infants with UCLP, even if outside air is taken into the nasal cavity, the air is discharged through the cleft. Therefore, intraoral pressure does not increase, and the infant cannot properly take in air. Furthermore, dry outside air and bacterial infections thicken the nasopharyngeal mucosa, which increases upper respiratory tract resistance [23,24,25]. Trindade et al. reported that vasoconstrictor administration reduced the thickening of the nasopharyngeal mucosa and improved SDB [26]. Therefore, the nasopharyngeal mucosa could be involved in upper respiratory tract resistance and sleep apnea status. Accordingly, SDB in infants with UCLP is attributable to the cleft between the hard and soft palates connecting the oral cavity, nasal cavity, and upper part of the nasopharyngeal cavity, which strongly inclines the nasal septum and thickens the nasopharyngeal mucosa. Additionally, this might exacerbate respiratory problems.

The Millard method of cheiloplasty is considered to have the most advantages with respect to restoring muscle function [27]. Reconstructing the orbicularis oris muscle and quadratus labii superioris (levator labii superioris and levator labii superioris alaeque nasi muscles) is crucial for the functional restoration of the cleft lip. Since the nostril bottom is formed before and after the muscular layer suture, the cleft part of the lip is closed, which allows the function of the orbicularis oris muscle to occur. This allows negative pressure to build in the oral cavity [28]. Therefore, the significant postoperative decrease in RDI observed in our study could result from cheiloplasty allowing lip closure, which shifts the breathing mode from oral to nasal and allows the nasal cavity functions of warming and humidification to occur. This could be attributed to decreased upper respiratory tract resistance caused by functional recovery and the decreased thickness of the nasopharyngeal mucosa [29].

The pushback method of palatoplasty closes the cleft part of the hard and soft palates and corrects the route of the levator veli palatini and palatopharyngeus muscles to form a muscle ring. This allows the elevation movement of the soft palate and nasopharyngeal contraction to occur [30,31]. This normalizes the nasopharyngeal closure function and respiratory airflow in the upper respiratory tract and improves swallowing and pronunciation [30,32]. Therefore, the significant decrease in RDI scores in our study could be attributed to the improvement in the environment of the upper respiratory tract.

In the present study, the mean RDI in infants with UCLP was 7.5 ± 4.6 events/h, which is indicative of mild sleep apnea. Notably, the mean RDI significantly decreased after cheiloplasty and palatoplasty compared with the baseline values, indicating a postoperative improvement in the breathing status. Interestingly, we observed temporary improvement in the RDI after cheiloplasty and worsening before palatoplasty. One possible explanation could be that the infants had been weaned during the period of 1 year and 6 months before palatoplasty. Further, during this period, their diet changed to solids. Food residues tend to stay in the nasal cavity, which increases the risk of bacterial infection. Therefore, the nasal mucosa may have been inflamed and thickened due to food residues and/or bacterial infection in the oral cavity or from the outside air. This could result in a higher RDI before palatoplasty than after cheiloplasty.

### 4.2. Pediatric SDB and RDI

Recent evidence shows greater health-care resource utilization, including longer hospital stay, for infants with obstructive sleep apnea or SDB. The management of SDB includes noninvasive ventilation or surgical interventions tailored for the patient. The screening of high-risk newborns should allow for early diagnosis and timely therapeutic intervention for this population. However, thresholds for diagnosing SDB and guiding and implementing treatment in neonates remain unclear. A collective effort is required to standardize the practice worldwide [33]. Therefore, in this study, the data for infants and young children were converted into data for children, and a judgment was made on the basis of the pediatric RDI.

### 4.3. Future Outlook

In the future, we aim to set SDB standards for infants and children based on the F-SAS Sensor^®^. In addition, we will conduct a similar study on infants with other cleft types to make comparisons according to cleft type.

## 5. Conclusions

We observed an improvement in SDB after cheiloplasty and palatoplasty in infants with UCLP. This suggests that both procedures may have a positive effect on the RDI and may reduce the adverse effects on growth and development in this population. Since the number of participants in this study was small, we plan to conduct a further verification of the results in a larger population.

## Figures and Tables

**Figure 1 diseases-13-00064-f001:**
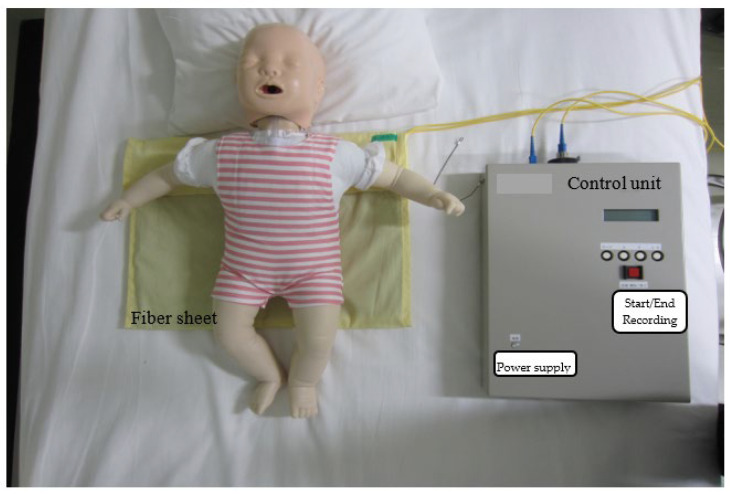
Schematic diagram of the F-SAS sensor^®^.

**Figure 2 diseases-13-00064-f002:**
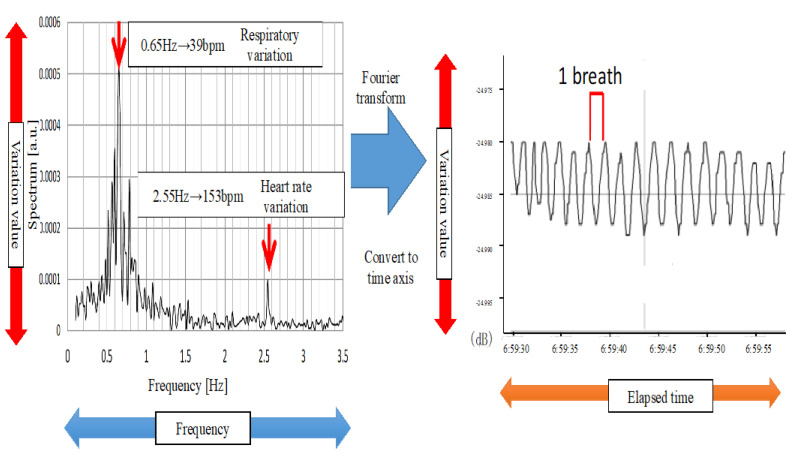
Measurement data and data after Fourier transformation.

**Figure 3 diseases-13-00064-f003:**
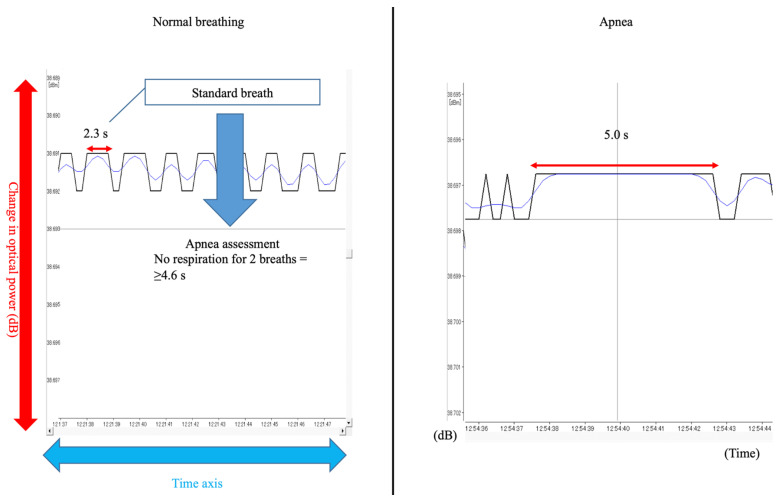
Examples of normal breathing and apnea waveforms.

**Figure 4 diseases-13-00064-f004:**
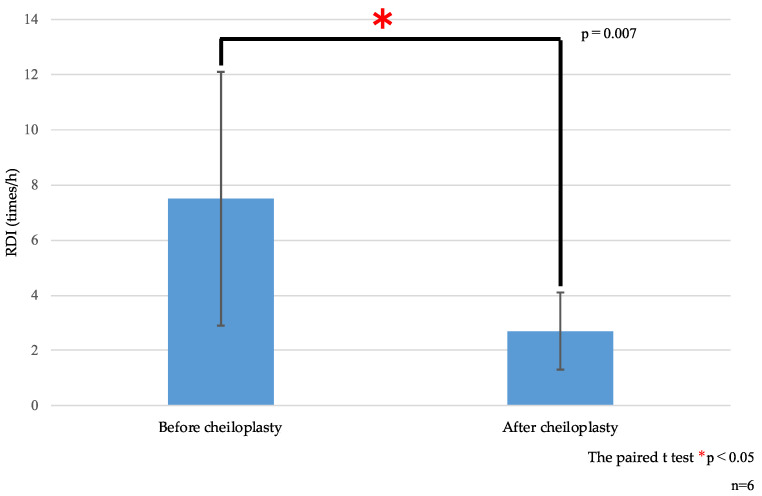
RDI before and after cheiloplasty. RDI, respiratory disturbance index.

**Figure 5 diseases-13-00064-f005:**
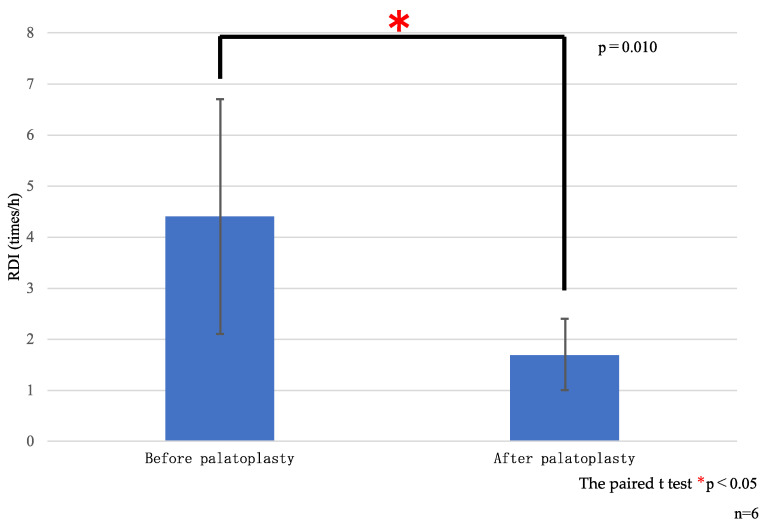
RDI before and after palatoplasty. RDI, respiratory disturbance index.

**Table 1 diseases-13-00064-t001:** Evaluation criteria for SDB severity through out-of-center sleep testing.

	Normal	Mild	Moderate	Severe
REI (times/h)	RDI < 5	5 ≤ RDI < 15	15 ≤ RDI < 30	30 ≤ RDI

RDI, respiratory disturbance index; calculated by dividing the total number of apneas and hypopneas during the test by the total test time.

**Table 2 diseases-13-00064-t002:** Participants’ characteristics according to the surgical technique.

	Target Infants			
	Cheiloplasty		Palatoplasty	
	Mean	S.D.	Mean	S.D.
Age (months)	6.0	1.9	20.0	4.2
Weight (kg)	6.7	0.8	11.3	1.0
Height (cm)	65.0	6.0	84.0	1.0
Kaup index (kg/m^2^)	16.0	2.0	16.0	1.5

S.D., standard deviation.

**Table 3 diseases-13-00064-t003:** Relationship with sleep apnea before and after cheiloplasty and palatoplasty.

	Preoperative RDI (Events/h)	Postoperative RDI (Events/h)	Preoperative vs. Postoperative
Cheiloplasty (n = 6)	7.5 ± 4.6	2.7 ± 1.4	0.007 *
Palatoplasty (n = 6)	4.4 ± 2.3	1.7 ± 0.7	0.010 *

RDI, respiratory disturbance index. * *p* < 0.5.

## Data Availability

The data presented in this study are available on request from the corresponding author. Data are not publicly available to protect patient privacy.

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
