# Peer review of "Examination of Respiratory Disturbance Index Before and After Cheiloplasty and Palatoplasty"

_diseases, 2025, doi:10.3390/diseases13030064_

Round 1
Reviewer 1 Report
Comments and Suggestions for Authors
I have read with interest the study. The use of the optical fiber-based sleep apnea sensor (Fiber-based Sleep Apnea Syndrome Sensor® ; hereinafter F-SAS Sensor® ) (AI Koabi, Japan) seems to be one of the main objectives of your study as it had not been previously used. this should be placed in you keywords and discussion. I do believe that your work is of great value.
Regarding the underlying reason for the temporary improvement in the RDI and worsening before palatoplasty, after lip formation, infants obtain passive immunity through breast milk [34]; This sentence doesn't have much value. Please improve
Author Response
Reviewer 1 Comments and Suggestions for Authors
I have read with interest the study. The use of the optical fiber-based sleep apnea sensor (Fiber-based Sleep Apnea Syndrome Sensor® ; hereinafter F-SAS Sensor® ) (AI Koabi, Japan) seems to be one of the main objectives of your study as it had not been previously used. this should be placed in you keywords and discussion. I do believe that your work is of great value.
Regarding the underlying reason for the temporary improvement in the RDI and worsening before palatoplasty, after lip formation, infants obtain passive immunity through breast milk [34]; This sentence doesn't have much value. Please improve.
Response 1: Thank you for bringing these points to our attention.
I added the optical fiber-based sleep apnea sensor (Fiber-based Sleep Apnea Syndrome Sensor® ; hereinafter F-SAS Sensor®) to the keywords.
The F-SAS sensor is mentioned in the discussion, but is there any need to add anything?
Regarding the underlying reason for the temporary improvement in the RDI and worsening before palatoplasty, after lip formation, infants obtain passive immunity through breast milk [34]; This sentence is deleted.
Reviewer 2 Report
Comments and Suggestions for Authors
In essence, it is my contention that this study has very limited significance and is not suitable for publication. The reasons for this conclusion are as follows: the authors studied the sleep alterations in children with cleft lip and palate after cleft lip and palate surgery. However, it is clear that cleft lip and palate repair is mandatory, and surgery will not be selectively performed or withheld just because of better, worse, or no change in the quality of sleep after surgery. It would be more beneficial to focus on how to more effectively address the sleep disorders that may accompany children with cleft lip and palate through surgery or related treatments. To this end, I propose to improve this study in the following way: to explore the improvement of sleep in patients with cleft lip and palate after different cleft lip surgery procedures or cleft palate surgery procedures.
Author Response
Reviewer 2 Comments and Suggestions for Authors
In essence, it is my contention that this study has very limited significance and is not suitable for publication. The reasons for this conclusion are as follows: the authors studied the sleep alterations in children with cleft lip and palate after cleft lip and palate surgery. However, it is clear that cleft lip and palate repair is mandatory, and surgery will not be selectively performed or withheld just because of better, worse, or no change in the quality of sleep after surgery. It would be more beneficial to focus on how to more effectively address the sleep disorders that may accompany children with cleft lip and palate through surgery or related treatments. To this end, I propose to improve this study in the following way: to explore the improvement of sleep in patients with cleft lip and palate after different cleft lip surgery procedures or cleft palate surgery procedures.
Response 2: Thank you for bringing these points to our attention.
This was merely a pilot study, and there was no detailed data available on measuring sleep dynamics after cheiloplasty and palate repair, so we conducted this research with the belief that observing sleep dynamics after cheiloplasty and palate repair would be an important step in evaluating the effectiveness of cleft lip and palate treatment.
We will continue to conduct research in accordance with the reviewer's comments in the future. We hope you will understand that we used this method this time.
Reviewer 3 Report
Comments and Suggestions for Authors
The authors performed a prospective cohort study consisting of 12 sample to assess the airway resistant after lip and palate repair among infants. The suggestions from this reviewer are listed below:
1. The abstracts need to be re-written to make it more understandable. Some sentences are difficult to understand.
2. In line 61-64, the authors mentioned about cleft lip and palate epidemiology then suddenly stated the aim. Seems out of place there. Would suggest the author to give a preamble on the relationship of cleft lip/palate with SDB from previous publication and add it there
3. Line 74, excluded or included?
4. Add the N value for each column in table 2
5. I would suggest to remove the subheading 4.1 and 4.2 as it does not help much in helping navigating the discussion
6. Line 239 to 247 explanation on food residue effect to airway seems speculative. Was the sample that underwent lip repair and palate repair the same subjects? If it is not that that itself (different subjects) could be the reasons of different RDI between post lip repair and pre-palate repair
7. Add a paragraph on imitation. Especially discuss regarding the reliability if the RDI assessment used in this study
8. Overall the English used need a significant revision preferable by a native English speakers. A lot of the sentence are confusing and difficult to understand. .
Comments on the Quality of English Language
Needs significant changes as some sentence cannot be understood
Author Response
The authors performed a prospective cohort study consisting of 12 sample to assess the airway resistant after lip and palate repair among infants. The suggestions from this reviewer are listed below:
- The abstracts need to be re-written to make it more understandable. Some sentences are difficult to understand.
Response 1: Thank you for bringing these points to our attention. The abstracts has been rewritten.
- In line 61-64, the authors mentioned about cleft lip and palate epidemiology then suddenly stated the aim. Seems out of place there. Would suggest the author to give a preamble on the relationship of cleft lip/palate with SDB from previous publication and add it there.
Response 2: We removed the section describing the epidemiology of cleft lip and palate and added a preface to our previous publication on the relationship between cleft lip/palate and SDB.
- Line 74, excluded or included?
Response 3: Line 74 includes.
- Add the N value for each column in table 2
Response 4: The N values are listed in Table 2.
- I would suggest to remove the subheading 4.1 and 4.2 as it does not help much in helping navigating the discussion
Response 5: Thank you for pointing this out, but I would like to leave it as is this time.
- Line 239 to 247 explanation on food residue effect to airway seems speculative. Was the sample that underwent lip repair and palate repair the same subjects? If it is not that that itself (different subjects) could be the reasons of different RDI between post lip repair and pre-palate repair
Response 6: Lines 239 to 249 will be deleted because there is data recorded on the same subjects who underwent both lip repair and palatal repair and data recorded on different subjects.
- Add a paragraph on imitation. Especially discuss regarding the reliability if the RDI assessment used in this study
Response 7: This is described in subheading 4.2.
- Overall the English used need a significant revision preferable by a native English speakers. A lot of the sentence are confusing and difficult to understand. .
Response 8: It has been corrected by a native English speakers.
Round 2
Reviewer 2 Report
Comments and Suggestions for Authors
Reviewer comment: Major revision
The article examines the effects of cheiloplasty and palatoplasty on sleep apnea in infants with unilateral cleft lip and palate (UCLP). It is a prospective cohort study involving 12 infants who underwent these surgical procedures. The respiratory disturbance index (RDI) was used to evaluate sleep-disordered breathing before and after surgery, with data collected using a fiber-based sleep apnea sensor. The results suggest that cheiloplasty and palatoplasty significantly improve sleep-disordered breathing in infants with UCLP. This improvement may have positive effects on their physical growth and development. The author's study is meaningful to the field of cleft lip and palate research; however, the following are some of my comments and concerns:
Abstract
1. In the abstract, the exact p values should be reported.
2. Your study has only one experimental group and no control group, so it can't be called a cohort study, it should be a single-arm trial or a before-after trial
Introduction
3. The author's introduction is very well written, accurate and concise, showing a good balance of ideas and concepts, and explaining the background and rationale of OSA and cleft lip and palate.
Methods
4. I think the authors' inclusion and exclusion criteria should be more clearly listed. In addition, were patients with the presence of other craniofacial syndromes besides cleft lip and palate included?
5. Your patients underwent surgery in 2 different hospitals. Is it one or more surgeons? Are their experience levels consistent?
6. Sleep respiratory status was monitored between 1 and 3 days post-surgery. However, the rationale for not employing a standardized and consistent monitoring interval requires further clarification.
7. You are using the wrong statistical method. The paired t test or wilcoxon signed-rank test should be used instead of the mann-Whitney U test for the comparison of pre-and post-treatment data.
Discussion
8. You did not calculate your sample size in your study, so a major limitation of this study is that the small sample may not reflect the improvement in sleep breathing after surgery in the overall cleft lip and palate population.
Conclusion
9. It is recommended to rewrite the conclusions, since the generalizations of your study to the broader pediatric population require further validation.
Author Response
Thank you for bringing these points to our attention.
Comment 1: In the abstract, the exact p values should be reported.
Reply 1: p-values added.
Comment 2: Your study has only one experimental group and no control group, so it can't be called a cohort study, it should be a single-arm trial or a before-after trial
Reply 2: This was changed to before-and-after study.
Comment 3: The author's introduction is very well written, accurate and concise, showing a good balance of ideas and concepts, and explaining the background and rationale of OSA and cleft lip and palate.
Reply 3: Only the English configuration has been changed.
Comment 4: I think the authors' inclusion and exclusion criteria should be more clearly listed.In addition, were patients with the presence of other craniofacial syndromes besides cleft lip and palate included?
Reply 4: Infants who could not undergo sleep breathing tests due to body movement or other reasons, those with other craniofacial syndromes in addition to UCLP, and those with sleeping time < 1 h were excluded.I rewrote it.
Comment 5: Your patientsunderwent surgery in 2 different hospitals. Is it one or more surgeons? Are their experience levels consistent?
Reply 5: All surgeons had similar experience levels.
Comment 6: Sleep respiratory status was monitored between 1 and 3 days post-surgery. However, the rationale for not employing a standardized and consistent monitoring interval requires further clarification.
Reply 6: Since the subjects are infants and young children, it cannot be helped that the periods differ like this.
Comment 7: You are using the wrong statistical method. The paired t test or wilcoxon signed-rank test should be used instead of the mann-Whitney U test for the comparison of pre-and post-treatment data.
Reply 7: Statistics were again performed using t-test.
Comment 8: You did not calculate your sample size in your study, so a major limitation of this study is that the small sample may not reflect the improvement in sleep breathing after surgery in the overall cleft lip and palate population.
Reply 8: Although the sample size is small this time, we believe that we can show a trend because we are targeting rare cases and there is an advantage.
Comment 9: It is recommended to rewrite the conclusions, since the generalizations of your study to the broader pediatric population require further validation.
Reply 9: I did a rewrite.
Reviewer 3 Report
Comments and Suggestions for Authors
This reviewer is in the opinion that the revised version has similar issues as the previous submitted manuscripts. Nothing much has been improved. The english written is still not to easy to follow and some corrections was ignored.
Comments on the Quality of English Language
I am not sure about the english revision made as not much changes in the writing. Maybe there is a need to get service of certified proofreader
Author Response
Thank you for bringing these points to our attention.
The text was then revised again by a certified proofreader service.
Round 3
Reviewer 2 Report
Comments and Suggestions for Authors
The author has successfully addressed all the concerns I raised, and I would like to express my gratitude for their efforts.